# One-year clinical outcomes of patients with versus without acute coronary syndrome with 3-month duration of dual antiplatelet therapy after everolimus-eluting stent implantation

**Masahiro Natsuaki[1], Takeshi Morimoto[2], Erika Yamamoto[3], Hirotoshi Watanabe[3], Yutaka Furukawa[4], Mitsuru Abe[5], Koichi Nakao[6], Tetsuya Ishikawa[7], Kazuya Kawai[8], Kei Yunoki[9], Shogo Shimizu[10], Masaharu Akao[5], Shinji Miki[11], Masashi Yamamoto[12], Hisayuki Okada[13], Kozo Hoshino[14], Kazushige Kadota[15], Yoshihiro Morino[16], Keiichi Igarashi Hanaoka[17], Kengo Tanabe[18], Ken Kozuma[19], Takeshi Kimura[3]\*, on behalf of the STOPDAPT trial investigators[¶]**

1 Department of Cardiovascular Medicine, Saga University, Saga, Japan, 2 Department of Clinical Epidemiology, Hyogo College of Medicine, Nishinomiya, Japan, 3 Department of Cardiovascular Medicine, Graduate School of Medicine, Kyoto University, Kyoto, Japan, 4 Department of Cardiovascular Medicine, Kobe City Medical Center General Hospital, Kobe, Japan, 5 Division of Cardiology, National Hospital Organization Kyoto Medical Center, Kyoto, Japan, 6 Division of Cardiology, Saiseikai Kumamoto Hospital, Kumamoto, Japan, 7 Division of Cardiology, Dokkyo Medical University Saitama Medical Center, Koshigaya, Japan, 8 Division of Cardiology, Chikamori Hospital, Kochi, Japan, 9 Division of Cardiology, Tsuyama Central Hospital, Tsuyama, Japan, 10 Division of Cardiology, Mashiko Hospital, Kawaguchi, Japan, 11 Division of Cardiology, Mitsubishi Kyoto Hospital, Kyoto, Japan, 12 Division of Cardiology, Kimitsu Chuo Hospital, Kimitsu, Japan, 13 Division of Cardiology, Seirei Hamamatsu General Hospital, Hamamatsu, Japan, 14 Division of Cardiology, Nagai Hospital, Tsu, Japan, 15 Department of Cardiology, Kurashiki Central Hospital, Kurashiki, Japan, 16 Division of Cardiology, Iwate Medical University Hospital, Morioka, Japan, 17 Division of Cardiology, Hanaokaseishu Memorial Cardiovascular Clinic, Sapporo, Japan, 18 Division of Cardiology, Mitsui Memorial Hospital, Tokyo, Japan, 19 Division of Cardiology, Teikyo University Hospital, Tokyo, Japan

¶ Membership listed in the acknowledgments.
* taketaka@kuhp.kyoto-u.ac.jp

**Data Availability Statement:** All relevant data are within the paper, its Supporting Information files

## Abstract

There has been no previous prospective study evaluating 3-month dual antiplatelet therapy (DAPT) after cobalt-chromium everolimus-eluting stent (CoCr-EES) implantation in patients with acute coronary syndrome (ACS). The STOPDAPT trial is a prospective multi-center single-arm study evaluating 3-month DAPT duration in all-comer population after CoCr-EES implantation. Among 1525 study patients enrolled from 58 Japanese centers, the present study compared the 1-year clinical outcomes between ACS patients (N = 487) and stable coronary artery disease (CAD) patients (N = 1038). In the ACS group, 228 patients (47%) had unstable angina and 259 patients (53%) had myocardial infarction. The primary endpoint was a composite of cardiovascular death, myocardial infarction, stroke, definite stent thrombosis (ST) and TIMI major/minor bleeding. Thienopyridine was discontinued within 4-month in 455 patients (94.0%) in the ACS group and 977 patients (94.3%) in the stable CAD group. Cumulative 1-year incidence of and the adjusted risk for the primary endpoint were not significantly different between the ACS and stable CAD groups (2.3% vs. 3.0%, P = 0.42, and HR 0.94, 95%CI 0.44–1.87, P = 0.87). In the 3-month landmark analysis,

and the Harvard Dataverse Databank with the following DOI: 10.7910/DVN/T4KPQI.

**Funding:** Abbott Vascular is the funding source of this study. The study sponsor was involved in the discussion on the study design for the main analysis. However, patient enrollment, data collection, statistical analysis, and manuscript preparation were conducted independent of the study sponsor.

**Competing interests:** Takeshi Kimura, Keiichi Igarashi, Kazushige Kadota, Kengo Tanabe, Yoshihiro Morino, and Ken Kozuma were advisory board members of Abbott Vascular. This does not alter our adherence to PLOS ONE policies on sharing data and materials.

cumulative incidence of the primary endpoint was also not significantly different between the ACS and stable CAD groups (1.3% vs. 2.4%, P = 0.16). There was no definite/probable ST through 1-year in both groups. In the propensity matched analysis, the cumulative 1-year incidence of the primary endpoint were similar between the ACS and stable CAD groups (2.3% versus 2.1%, P = 0.82). In conclusion, stopping DAPT at 3 months after CoCr-EES implantation in patients with ACS including 47% of unstable angina was as safe as that in patients with stable CAD.

## Introduction

The current American Heart Association (AHA) and European Society of Cardiology (ESC) guidelines recommend 6-month dual antiplatelet therapy (DAPT) after drug-eluting stent (DES) implantation in patients with stable coronary artery disease (CAD).[1,2] On the other hand, DAPT has been recommended for at least 1 year in patients with acute coronary syndrome (ACS), irrespective of the revascularization strategies and stent types. However, due to the paucity of dedicated randomized trials or prospective trial evaluating short DAPT in patients with ACS, the optimal duration of DAPT after DES implantation in ACS is still a matter of debate.

We previously reported the favorable outcomes of those patients treated with 3-month DAPT after cobalt-chromium everolimus-eluting stent (CoCr-EES) in the STOPDAPT (ShorT and OPtimal duration of Dual AntiPlatelet Therapy after everolimus-eluting cobalt-chromium stent) trial as compared with those enrolled in the historical control of RESET (Randomized Evaluation of Sirolimus-eluting versus Everolimus-eluting stent Trial) study, in which nearly 90% of patients had continued DAPT at 1-year. [3,4] The STOPDAPT trial included a substantial proportion of patients with ACS. Therefore, we sought to evaluate the safety of 3-month DAPT duration after CoCr-EES implantation in those with ACS as compared with those with stable CAD.

## Methods

### Study population

STOPDAPT trial is a prospective multi-center single-arm trial enrolling patients who agreed to follow the 3-month DAPT protocol (discontinuation of clopidogrel at 2- to 4-month and aspirin monotherapy thereafter) after successful CoCr-EES implantation in all-comer population.[3] Patients who underwent successful percutaneous coronary intervention (PCI) using CoCr-EES were to be enrolled, if the physicians in charge judged the patient to be eligible for the study evaluating 3-month DAPT duration. Patients who had previous history of PCI using DES other than CoCr-EES were excluded.

Between September 2012 and October 2013, 6070 patients underwent PCI using CoCr-EES in 58 Japanese centers (List A in S1 Appendix). We excluded 2490 patients who were previously treated with DES other than CoCr-EES. Among 3580 eligible patients, 1526 patients (43%) were enrolled in this study. Excluding 1 patient who withdrew consent for study participation, 1525 patients constituted the current study population. Among 1525 patients, 487 patients presented as ACS and 1038 patients had stable CAD (Fig 1). Complete 1-year clinical follow-up was achieved in 1519 patients (99.6%). We compared the clinical outcomes between ACS and stable CAD patients.

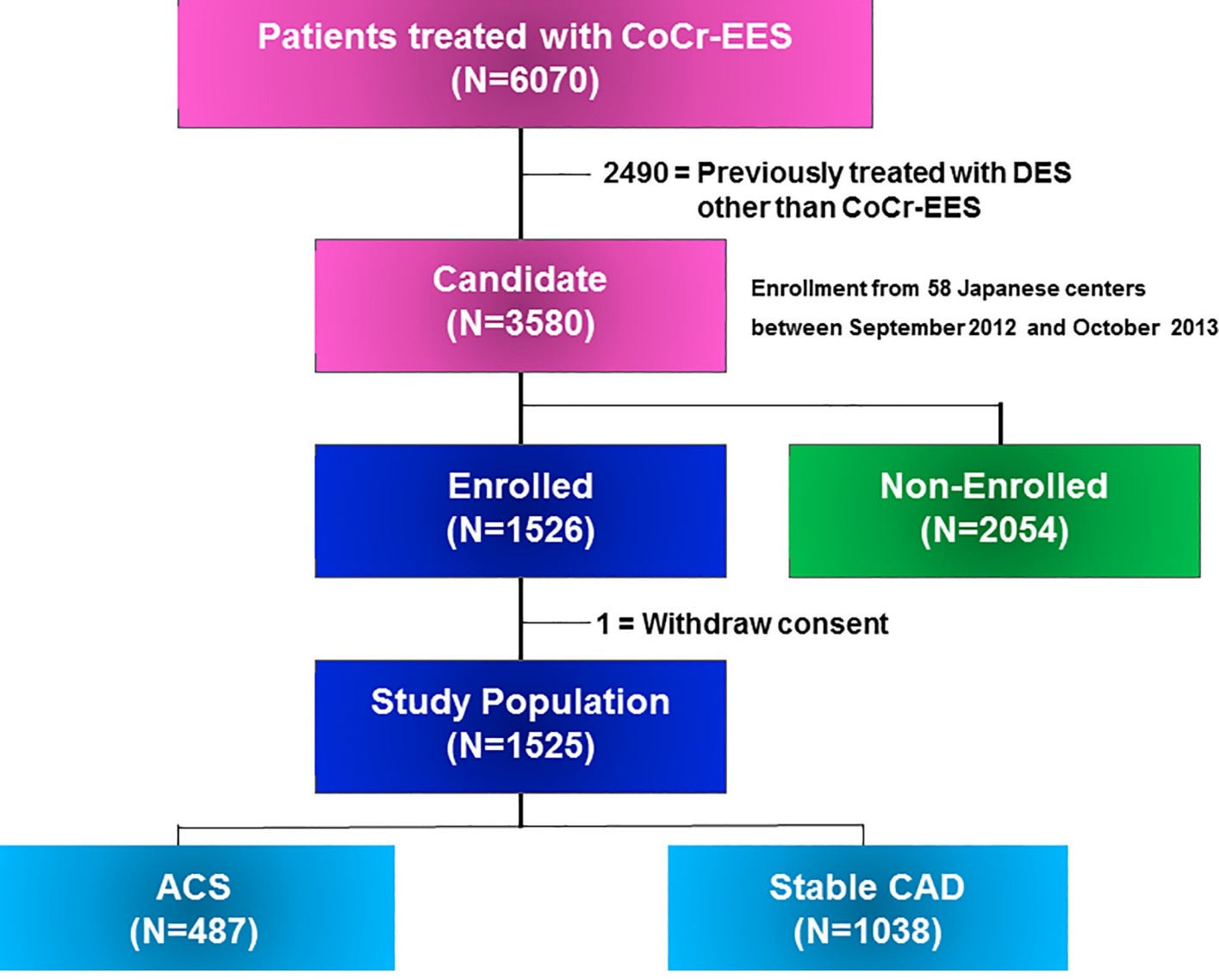

**Fig 1. Study flow chart.** CoCr-EES, Cobalt-chromium everolimus-eluting stent; DES, drug-eluting stent; ACS, acute coronary syndrome; CAD, coronary artery disease.

As a historical control group, we selected the CoCr-EES group in the RESET trial (a randomized controlled trial comparing CoCr-EES with sirolimus-eluting stent conducted by the same study group in 2010), where nearly 90% of patients had continued DAPT at 1-year.[4] The eligibility criteria of the RESET was comparable to that of the STOPDAPT except for the inclusion of patients with previous DES implantation in the RESET. Among 1597 patients in the CoCr-EES group in the RESET, 38 patients with in-hospital primary endpoint events were excluded from the historical control group in this study, because patients in the STOPDAPT were enrolled after completion of successful PCI. A total of 1559 patients were selected as a historical control group.[3]

## Ethics

The research protocol of the STOPDAPT trial (S1 Protocol) was approved by the Institutional Review Board in Kyoto University and by the local ethics committees in all of 58 participating medical centers (List A in S1 Appendix) (ClinicalTrials.gov: NCT 01659034). Written informed consent was obtained from all the study patients.

## Procedures

Antiplatelet regimen included aspirin (≥81mg daily) indefinitely and thienopyridine (75mg clopidogrel daily) for 3-month after stent implantation. Ticlopidine 200 mg/day was only allowed for those who did not tolerate clopidogrel. Patients were instructed to discontinue thienopyridine at 3-month hospital visit. Acceptable time window for the discontinuation of thienopyridine therapy was within ±1 month. DAPT duration in patients with oral anticoagulants was the same as the rest of the population. Status of antiplatelet therapy was evaluated throughout the follow-up period as previously described.[5] Persistent discontinuation of thienopyridine was defined as withdrawal lasting for at least 2 months.[5]

## Endpoints and definitions

ACS included those patients who presented as ST-segment elevation myocardial infarction (STEMI), non-ST-segment elevation myocardial infarction (NSTEMI), and unstable angina (UA) at the time of index PCI. The remaining study patients other than ACS were regarded as having stable CAD. STEMI was defined as those with an increase of cardiac biomarker with persistent ST-segment elevation or new Q-wave in electrocardiogram. NSTEMI was defined as those with an increase of cardiac biomarker without electrocardiogram changes of STEMI. High-sensitivity troponin or CK-MB value > upper reference limit were diagnosed as MI in the all participating centers. UA was defined as those with anginal pain of Braunwald class I-III with no increase of a cardiac biomarker.[6] The Global Registry of Acute Coronary Events (GRACE) score was calculated in patients with ACS.[7] The primary endpoint in this trial was a composite of cardiovascular death, myocardial infarction (MI), stroke, definite stent thrombosis (ST) and Thrombolysis in Myocardial Infarction (TIMI) major/minor bleeding at 1 year. Primary endpoint events were adjudicated by the independent clinical event committee (List B in S1 Appendix). Major secondary endpoints were TIMI major/minor bleeding and a composite of cardiovascular death, MI, stroke or definite ST at 1 year. Other secondary endpoints included death, MI, stroke, possible/probable/definite ST, bleeding events defined by TIMI or Global Utilization of Streptokinase and Tissue plasminogen activator for Occluded coronary arteries (GUSTO) criteria,[8,9] target-lesion revascularization (TLR), target-vessel revascularization (TVR), coronary artery bypass grafting and any coronary revascularization.

Death was regarded as cardiac in origin unless obvious non-cardiac causes could be identified. MI and ST were defined according to the Academic Research Consortium definitions. [10] Stroke during follow-up was defined as ischemic or hemorrhagic stroke requiring hospitalization with symptoms lasting >24 hours. TLR was defined as either PCI or coronary artery bypass grafting due to restenosis or thrombosis of the target lesion that included the proximal and distal edge segments as well as the ostium of the side branches. Patients with planned staged PCI were enrolled in this study after the completion of all the planned coronary revascularization procedures. Therefore, scheduled staged PCI procedures were not included in any coronary revascularization during follow-up.

## Data collection and follow-up

Demographic, angiographic, and procedural data were collected from hospital charts or data-bases in each participating center according to the pre-specified definitions by experienced clinical research coordinators in the participating centers (List B in S1 Appendix) or in the study management center (List B in S1 Appendix). Follow-up data on the clinical events were collected from the hospital charts in the participating centers (74%), letters to patients (20%), and telephone call to referring physicians (8.4%).

## Statistical analysis

Categorical variables were presented as number and percentage, and were compared with the chi-square test or the Fisher's exact test. Continuous variables were expressed as mean value ± SD or median with inter-quartile range, and were compared using the Student's t test or Wilcoxon rank sum test based on their distributions. As the main analysis in the present study, clinical outcomes were compared between the ACS patients and the stable CAD patients in the STOPDAPT trial. Cumulative incidence was estimated by the Kaplan-Meier method and differences were assessed with the log-rank test. To evaluate the events beyond 3-month, we also conducted the landmark analyses at 3-month. Those patients who had the individual endpoint events before 3-month were excluded in the landmark analyses. Due to the presence of differences in baseline characteristics between the 2 groups, we used multivariable Cox pro-portional hazard models to estimate the risk of patients with ACS relative to those with stable CAD for the primary endpoint. In the multivariable analysis, we chose 7 clinically relevant fac-tors indicated in Table 1 as the risk adjusting variables (age > = 75 years, diabetes, hemodialy-sis, atrial fibrillation, prior MI, peripheral vascular disease, and statins use). The continuous variables were dichotomized by clinically meaningful reference values or median values. The presentation (ACS or stable CAD), the 7 risk-adjusting variables and random effect of center were simultaneously included in the Cox proportional hazard model. Proportional hazard assumptions for the risk-adjusting variables were assessed on the plots of log (time) versus log [-log (survival)] stratified by the variable. The assumptions were verified to be acceptable for all the variables. The effect of ACS relative to stable CAD for the primary endpoint was expressed as hazard ratios (HR) and their 95% confidence intervals (CI). As the sensitivity analyses, we conducted propensity matched analysis. The propensity score was calculated from the 7 risk-adjusting variables. Using the propensity score, patients in the stable CAD group were randomly matched to ACS patients using a greedy matching strategy. Sample size calculation for the main analysis of this study was previously described. [3]

Statistical analyses were conducted by a physician (Natsuaki M) and a statistician (Morimoto T) with the use of JMP 10.0 software. We used 2-sided P values <0.05 as statistically significant.

## Results

### Baseline characteristics: Enrolled versus non-enrolled patients in the STOPDAPT

Baseline characteristics were significantly different in several aspects between the enrolled and non-enrolled patients (S1 Table). AMI presentation were more prevalent in the non-enrolled group, while stable CAD were more often found in the enrolled group.

### Baseline characteristics: ACS versus stable CAD

In ACS group, 44% of patients presented as STEMI, 9% of patients presented as NSTEMI and 47% of patients had UA. Baseline characteristics were significantly different in several aspects

**Table 1. Baseline characteristics: ACS versus stable CAD.**

| | ACS | Stable CAD | P Value |
|---|---|---|---|
| | 487 patients | 1038 patients | |
| | 555 lesions | 1283 lesions | |
| **Clinical characteristics** | | | |
| Age–years | 69.0±12.3 | 70.5±9.7 | 0.009 |
| Age > = 75 years * | 176 (36%) | 394 (38%) | 0.49 |
| Men | 361 (74%) | 756 (73%) | 0.59 |
| Body mass index | 24.0±3.6 | 24.2±3.5 | 0.21 |
| Coexisting condition | | | |
| Hypertension | 406 (83%) | 855 (82%) | 0.63 |
| Diabetes mellitus * | 162 (33%) | 442 (43%) | 0.0005 |
| Insulin-treated diabetes | 18 (3.7%) | 101 (9.7%) | <0.0001 |
| Treated with oral medication only | 91 (19%) | 269 (26%) | 0.002 |
| Treated with diet therapy only | 53 (11%) | 72 (6.9%) | 0.01 |
| Dyslipidemia | 386 (79%) | 823 (79%) | 0.99 |
| ESRD (eGFR<30 mL/min/1.73m$^2$) not on HD | 16 (3.3%) | 19/1034 (1.8%) | 0.09 |
| HD * | 8 (1.6%) | 48 (4.6%) | 0.002 |
| Atrial fibrillation * | 39 (8.0%) | 133 (13%) | 0.005 |
| Anemia (Hemoglobin <11.0 g/dL) | 76 (16%) | 165 (16%) | 0.88 |
| Cardiac risk factor | | | |
| Current smoker | 142 (29%) | 173 (17%) | <0.0001 |
| Family history of coronary artery disease | 58 (12%) | 134 (13%) | 0.58 |
| Prior myocardial infarction * | 38 (7.8%) | 229 (22%) | <0.0001 |
| Prior Stroke | 49 (10%) | 119 (11%) | 0.41 |
| Heart failure | 58 (12%) | 147 (14%) | 0.23 |
| Peripheral vascular disease * | 19 (3.9%) | 123 (12%) | <0.0001 |
| Prior percutaneous coronary intervention | 80 (16%) | 388 (37%) | <0.0001 |
| Prior coronary-artery bypass grafting | 6 (1.2%) | 35 (3.4%) | 0.01 |
| Clinical characteristics | | | |
| Clinical presentation | | | |
| Unstable angina | 228 (47%) | | |
| Acute myocardial infarction | 259 (53%) | | |
| STEMI | 215 (44%) | | |
| NSTEMI | 44 (9.0%) | | |
| GRACE score | | | |
| Unstable angina | 109.4±27.0 | | |
| STEMI | 159.6±32.1 | | |
| NSTEMI | 123.8±32.5 | | |
| Left ventricular ejection fraction <30% | 4/419 (1.0%) | 13/896 (1.5%) | 0.45 |
| Multivessel disease | 141 (29%) | 437 (42%) | <0.0001 |
| Target vessel location | | | |
| Left main coronary artery | 4 (0.8%) | 13 (1.3%) | 0.44 |
| Left anterior descending coronary artery | 299 (61%) | 567 (55%) | 0.01 |
| Left circumflex coronary artery | 85 (17%) | 276 (27%) | <0.0001 |
| Right coronary artery | 125 (26%) | 280 (27%) | 0.59 |
| Bypass graft | 1 (0.2%) | 3 (0.3%) | 0.76 |
| Complexity of coronary artery disease | | | |
| No. of treated lesions per patient | 1.14±0.42 | 1.24±0.5 | <0.0001 |

(*Continued*)

**Table 1.** (Continued)

| | ACS | Stable CAD | P Value |
|---|---|---|---|
| | **487 patients** | **1038 patients** | |
| | **555 lesions** | **1283 lesions** | |
| Medications | | | |
| Aspirin | 487 (100%) | 1037 (99.9%) | 0.38 |
| Thienopyridines | 487 (100%) | 1035 (99.7%) | 0.13 |
| Clopidogrel | 485 (99.6%) | 1023 (98.8%) | 0.13 |
| Ticlopidine | 2 (0.4%) | 12 (1.2%) | |
| Statins * | 438 (90%) | 785 (76%) | <0.0001 |
| Strong statins** | 407 (84%) | 706 (68%) | <0.0001 |
| Maximum approved doses of strong statin | 7 (1.4%) | 20 (1.9%) | 0.49 |
| B-blockers | 256 (53%) | 364 (35%) | <0.0001 |
| ACE-I/ARB | 334 (69%) | 605 (58%) | 0.0001 |
| Calcium-channel blockers | 164 (34%) | 511 (49%) | <0.0001 |
| Nitrates | 65 (13%) | 154 (15%) | 0.44 |
| Anticoaglants | 45 (9.2%) | 123 (12%) | 0.12 |
| Warfarin | 38 (7.8%) | 87 (8.4%) | 0.7 |
| Dabigatran | 5 (1.0%) | 29 (2.8%) | 0.02 |
| Rivaroxaban | 2 (0.4%) | 7 (0.7%) | 0.52 |
| **Lesion and procedural characteristics** | | | |
| Before index procedure | | | |
| Chronic total occlusion | 7 (1.4%) | 65 (6.3%) | <0.0001 |
| Culprit for STEMI | 202 (41%) | | |
| Bifurcation | 105 (22%) | 212 (20%) | 0.61 |
| After index procedure | | | |
| No. of stents used per patient | 1.27±0.56 | 1.42±0.68 | <0.0001 |
| Total stent length per patient—mm | 30.7±18.0 | 33.9±22.1 | 0.09 |
| Multivessel treatment | 28 (5.8%) | 102 (9.8%) | 0.006 |

Values are expressed as mean ± SD or number (%).

* Risk-adjusting variables selected for multivariable analysis.

** Atorvastatin, pitavastatin and rosuvastatin were considered to be strong statins.

ACS = acute coronary syndrome; CAD = coronary artery disease; ESRD = end stage renal disease; eGFR = estimated glomerular filtration rate; HD = hemodialysis; STEMI = ST-segment elevation myocardial infarction; NSTEMI = non-ST-segment elevation myocardial infarction; The Global Registry of Acute Coronary Events (GRACE); ACE-I = angiotensin converting enzyme inhibitors; ARB = angiotensin II receptor blockers.

between the ACS and stable CAD groups (Table 1). Patients in the ACS group were significantly younger than those in the stable CAD group. The ACS group more often had current smoker, while the stable CAD group more often had diabetes, hemodialysis, atrial fibrillation, prior MI, peripheral vascular disease, prior PCI, prior coronary artery bypass grafting and multi-vessel disease. The ACS group more often included treatment of left anterior descending artery, while the stable CAD group more often included treatment of left circumflex artery and chronic total occlusion. The stable CAD group had greater number of stents per patient, and longer total stent length per patient than the ACS group. Multi-vessel treatment was more often performed in the stable CAD group than in the ACS group. Regarding the medications at hospital discharge, statins, β-blockers and angiotensin converting enzyme inhibitors/angiotensin II receptor blockers were more often prescribed in the ACS group than in the non-ACS group. Maximum approved doses of strong statins were very infrequently implemented (Table 1).

### Discontinuation of thienopyridine

Thienopyridine was discontinued within 4-month in 455 patients (94.0%) in the ACS group and 977 patients (94.3%) in the stable CAD group. Cumulative 1-year incidence of persistent discontinuation of thienopyridine was 96.7% in the ACS group and 96.8% in the stable CAD group (P = 0.34) (Fig 2).

### Clinical outcomes through 1-year

Cumulative 1-year incidence of the primary endpoint was very low and not significantly different between the ACS and stable CAD groups (2.3% versus 3.0%, P = 0.42) (Fig 3 and Table 2). In the multivariable analysis, there was no excess risk of the ACS group relative to the stable CAD group for the primary endpoint (adjusted HR 0.94, 95% CI 0.44–1.87, P = 0.87) (Table 3). Regarding the major secondary endpoints, the cumulative incidences of TIMI major/minor bleeding and a composite of cardiovascular death, MI, stroke and definite ST were also not significantly different between the ACS and stable CAD groups. There was no definite/probable ST through 1-year after PCI in both groups (Fig 4).

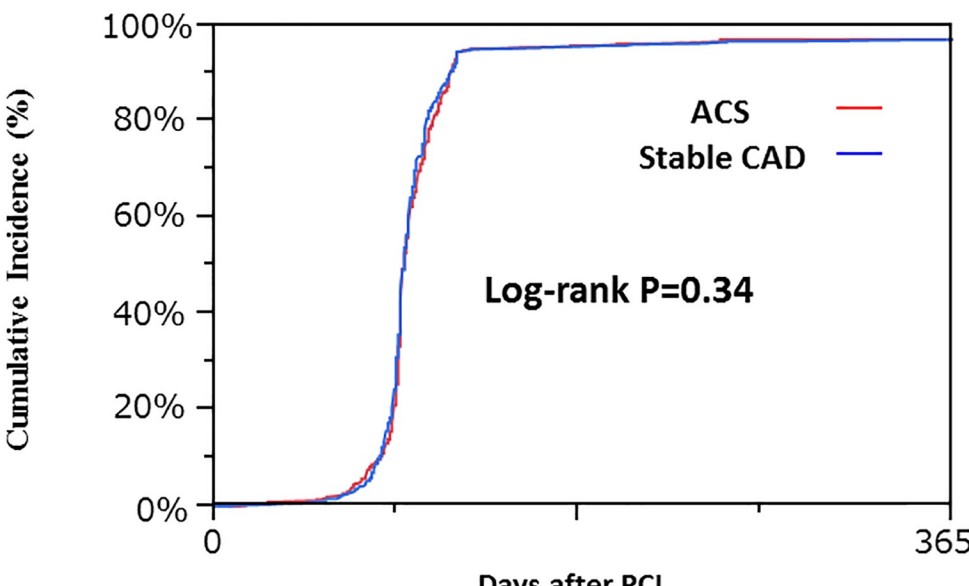

| Interval | 0 day | 30 days | 90 days | 120 days | 180 days | 240 days | 365 days |
|---|---|---|---|---|---|---|---|
| **ACS** | | | | | | | |
| N of patients with discontinuation | | 5 | 123 | 455 | 455 | 462 | 468 |
| N of patients at risk | 487 | 480 | 361 | 29 | 22 | 18 | 16 |
| Cumulative Incidence | | 1.0% | 25.4% | 94.0% | 95.5% | 96.3% | 96.7% |
| **Stable CAD** | | | | | | | |
| N of patients with discontinuation | | 6 | 320 | 977 | 989 | 996 | 1003 |
| N of patients at risk | 1038 | 1032 | 717 | 59 | 47 | 40 | 31 |
| Cumulative Incidence | | 0.6% | 30.9% | 94.3% | 95.5% | 96.1% | 96.8% |

**Fig 2. Cumulative incidence of persistent discontinuation of thienopyridine.** ACS, acute coronary syndrome; CAD, coronary artery disease; PCI, percutaneous coronary intervention.

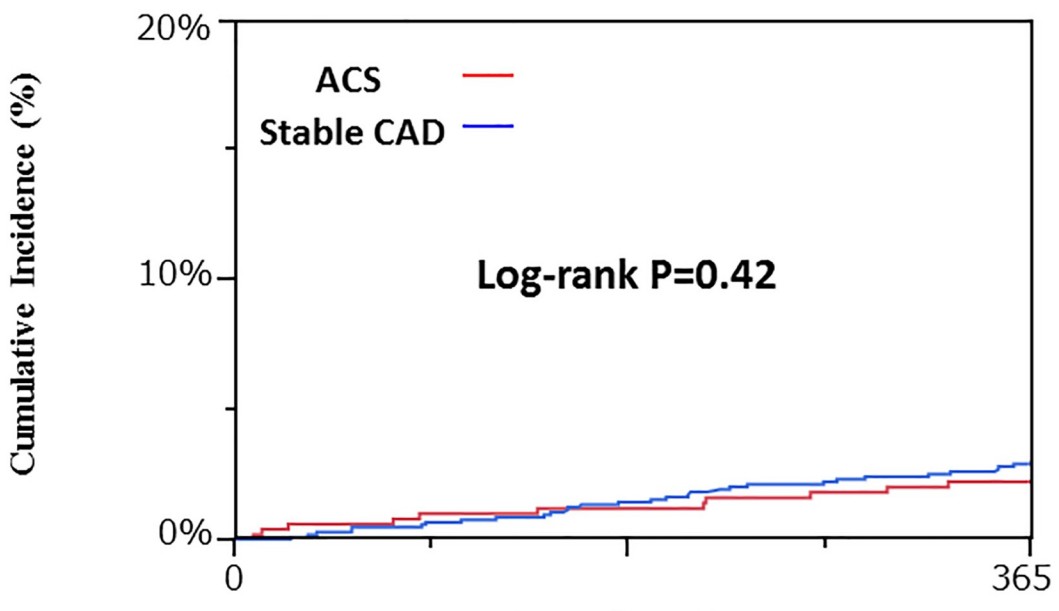

**Primary Endpoint**
**Cardiovascular death, MI, Stroke, Definite ST, and Bleeding**

| Interval | 0 day | 30 days | 180 days | 240 days | 365 days |
|---|---|---|---|---|---|
| **ACS** | | | | | |
| N of patients with at least 1 event | | 3 | 6 | 8 | 11 |
| N of patients at risk | 487 | 483 | 477 | 475 | 472 |
| Cumulative Incidence | | 0.6% | 1.2% | 1.7% | 2.3% |
| **Stable CAD** | | | | | |
| N of patients with at least 1 event | | 1 | 15 | 22 | 31 |
| N of patients at risk | 1038 | 1037 | 1013 | 1005 | 986 |
| Cumulative Incidence | | 0.1% | 1.5% | 2.1% | 3.0% |

**Fig 3. Cumulative incidence of the primary endpoint.** Primary endpoint was defined as a composite of cardiovascular death, MI, stroke, definite ST and TIMI major/minor bleeding. MI, myocardial infarction; ST, stent thrombosis; TIMI, Thrombolysis in Myocardial Infarction; ACS, acute coronary syndrome; CAD, coronary artery disease; PCI, percutaneous coronary intervention.

### Clinical outcomes between 3- and 12-month

Between 3- and 12-month, the cumulative incidence of the primary endpoint was not significantly different between the ACS and stable CAD groups (1.3% versus 2.4%, P = 0.16) (Fig 5 and Table 4). Regarding the major secondary endpoints, the cumulative incidences of TIMI major/minor bleeding and a composite of cardiovascular death, MI, stroke and definite ST were also not significantly different between the ACS and stable CAD groups. No patients had definite or probable ST between 3- and 12-month in both groups. Cumulative incidences of all-cause death, non-cardiac death and a composite of death or MI were significantly lower in the ACS group than in the stable CAD group (Table 4).

**Table 2. Clinical outcomes at 12-month.**

| | No. of patients with at least one event (Cumulative incidence) | | P Value |
|---|---|---|---|
| | ACS | Stable CAD | |
| | N = 487 | N = 1038 | |
| Primary Endpoint | 11 (2.3%) | 31 (3.0%) | 0.42 |
| Death | | | |
| All-cause | 5 (1.0%) | 25 (2.4%) | 0.07 |
| Cardiac death | 2 (0.4%) | 7 (0.7%) | 0.53 |
| Cardiovascular death | 2 (0.4%) | 8 (0.8%) | 0.42 |
| Non-cardiac death | 3 (0.6%) | 18 (1.7%) | 0.08 |
| Myocardial infarction | 1 (0.2%) | 3 (0.3%) | 0.76 |
| Stroke | | | |
| Any | 4 (0.8%) | 13 (1.3%) | 0.45 |
| Ischemic | 3 (0.6%) | 11 (1.1%) | 0.4 |
| Hemorrhagic | 2 (0.4%) | 2 (0.2%) | 0.44 |
| Bleeding | | | |
| TIMI major | 4 (0.8%) | 8 (0.8%) | 0.92 |
| TIMI minor/major | 6 (1.2%) | 9 (0.9%) | 0.5 |
| TIMI minimal/minor/major | 14 (2.9%) | 23 (2.2%) | 0.44 |
| GUSTO severe | 4 (0.8%) | 6 (0.6%) | 0.59 |
| GUSTO moderate/severe | 4 (0.8%) | 12 (1.2%) | 0.55 |
| Definite stent thrombosis | 0 (0%) | 0 (0%) | |
| Stent thrombosis | | | |
| Possible | 0 (0%) | 6 (0.6%) | 0.09 |
| Probable | 0 (0%) | 0 (0%) | |
| Definite or probable | 0 (0%) | 0 (0%) | |
| Definite, probable or possible | 0 (0%) | 6 (0.6%) | 0.09 |
| Death or myocardial infarction | 6 (1.2%) | 28 (2.7%) | 0.07 |
| Cardiovascular death or myocardial infarction | 3 (0.6%) | 11 (1.1%) | 0.4 |
| Cardiovascular death, MI or stroke | 7 (1.4%) | 24 (2.3%) | 0.26 |
| Cardiovascular death, MI, stroke or definite ST | 7 (1.4%) | 24 (2.3%) | 0.26 |
| Target-lesion revascularization | 8 (1.7%) | 22 (2.2%) | 0.52 |
| Target-vessel revascularization | 15 (3.1%) | 40 (3.9%) | 0.44 |
| Any coronary revascularization | 30 (6.2%) | 79 (7.7%) | 0.28 |
| Coronary-artery bypass grafting | 1 (0.2%) | 2 (0.2%) | 0.96 |

Values are expressed as number (%).

Cumulative incidence was estimated by Kaplan-Meier method.

ACS = acute coronary syndrome; CAD = coronary artery disease; TIMI = Thrombolysis in Myocardial Infarction; GUSTO = Global Utilization of Streptokinase and Tissue plasminogen activator for Occluded coronary arteries; MI = myocardial infarction; ST = stent thrombosis.

## Propensity matched analysis

As a sensitivity analysis, we compared the clinical outcomes between the ACS and stable CAD patients in the propensity matched cohort. Baseline characteristics were similar in terms of higher age, diabetes, hemodialysis, AF, prior MI, PVD and statin use between the 2 groups. Dyslipidemia, prior PCI and multivessel disease were more included in the stable CAD group, while current smoker were more often found in the ACS group (S2 Table). Cumulative 1-year

**Table 3. Unadjusted and adjusted risks of the ACS versus non-ACS and the effects of the risk-adjusting variables for the primary endpoint.**

| | Unadjusted | P value | Adjusted | P value |
|---|---|---|---|---|
| | HR (95%CI) | | HR (95%CI) | |
| ACS | 0.75 (0.36–1.46) | 0.41 | 0.94 (0.46–1.92) | 0.87 |
| Age > = 75 years | 2.53 (1.38–4.76) | 0.003 | 2.1 (1.11–3.94) | 0.02 |
| Diabetes | 0.76 (0.39–1.42) | 0.4 | 0.71 (0.37–1.36) | 0.3 |
| Hemodialysis | 2.04 (0.49–5.62) | 0.28 | 1.62 (0.49–5.36) | 0.43 |
| Atrial fibrillation | 2.53 (1.18–4.97) | 0.02 | 2.03 (0.98–4.19) | 0.06 |
| Prior myocardial infarction | 1.48 (0.69–2.91) | 0.29 | 1.54 (0.74–3.19) | 0.24 |
| Peripheral vascular disease | 3.14 (1.46–6.15) | 0.005 | 2.54 (1.23–5.28) | 0.01 |
| Statins use | 0.78 (0.4–1.67) | 0.5 | 0.9 (0.43–1.88) | 0.78 |

ACS = acute coronary syndrome, HR = hazard ratio, CI = confidence interval.

incidence of the primary endpoint were similar between the ACS and stable CAD groups (2.3% versus 2.1%, P = 0.82) (Fig 6 and S3 Table).

## Discussion

The main finding of the current study is that stopping DAPT at 3-month after CoCr-EES implantation in patients with ACS was as safe as that in patients with stable CAD.

The current guidelines recommend 6-month DAPT after DES implantation in patients with stable CAD.[1,2] However, there is a widely recognized notion that patients who presented with ACS have increased risk for subsequent atherothrombotic events as compared with those with stable CAD. Therefore, prolonged DAPT duration is recommended for more intensive prophylaxis for progressive atherothrombosis.[11,12] Current recommendation of at least 1 year DAPT duration in ACS patients is based on the assumption that increased plaque vulnerability in ACS patients would be stabilized up to 1 year after the index ACS event. However, we previously reported that patients with acute myocardial infarction (AMI) as compared with those without AMI have similar cardiovascular event risk beyond 3 months after PCI, suggesting that stabilization of plaque vulnerability in ACS patients might have been achieved earlier than the currently assumed time period of 1 year.[13] In the meta-analyses of trials comparing DAPT duration after PCI, prolonged DAPT as compared with short DAPT was associated with increased risk of major bleeding and with a signal of increased mortality. [14,15] Therefore, we have a reasonable rationale to explore shortening of the mandatory DAPT duration after DES implantation in ACS patients.

There is a scarcity of randomized trials exploring short DAPT in ACS patients. In the PCI-CURE (Clopidogrel in Unstable Angina to Prevent Recurrent Events) study, long-term administration of clopidogrel after PCI in patients with non-ST elevation ACS was associated with a lower rate of cardiovascular death, MI, or any revascularization.[16] However, the study was conducted almost 20 years ago, when the implementation of the secondary preventive measures was much different from that in contemporary clinical practice. Furthermore, the study design of the PCI-CURE study might be flawed, because pretreatment of clopidogrel before PCI was implemented only in those patients assigned to long-term clopidogrel, but not in those patients assigned to placebo. In the SMART-DATE (Six-month versus 12-month or longer dual antiplatelet therapy after percutaneous coronary intervention in patients with acute coronary syndromes) trial, 6-month DAPT was non-inferior to 12-month DAPT in patients with ACS with new-generation DES in terms of a composite of all-cause death, MI, or stroke at 18 months (4.7% versus 4.2%, P for non-inferiority = 0.03).[17] In the DAPT-STEMI (Six

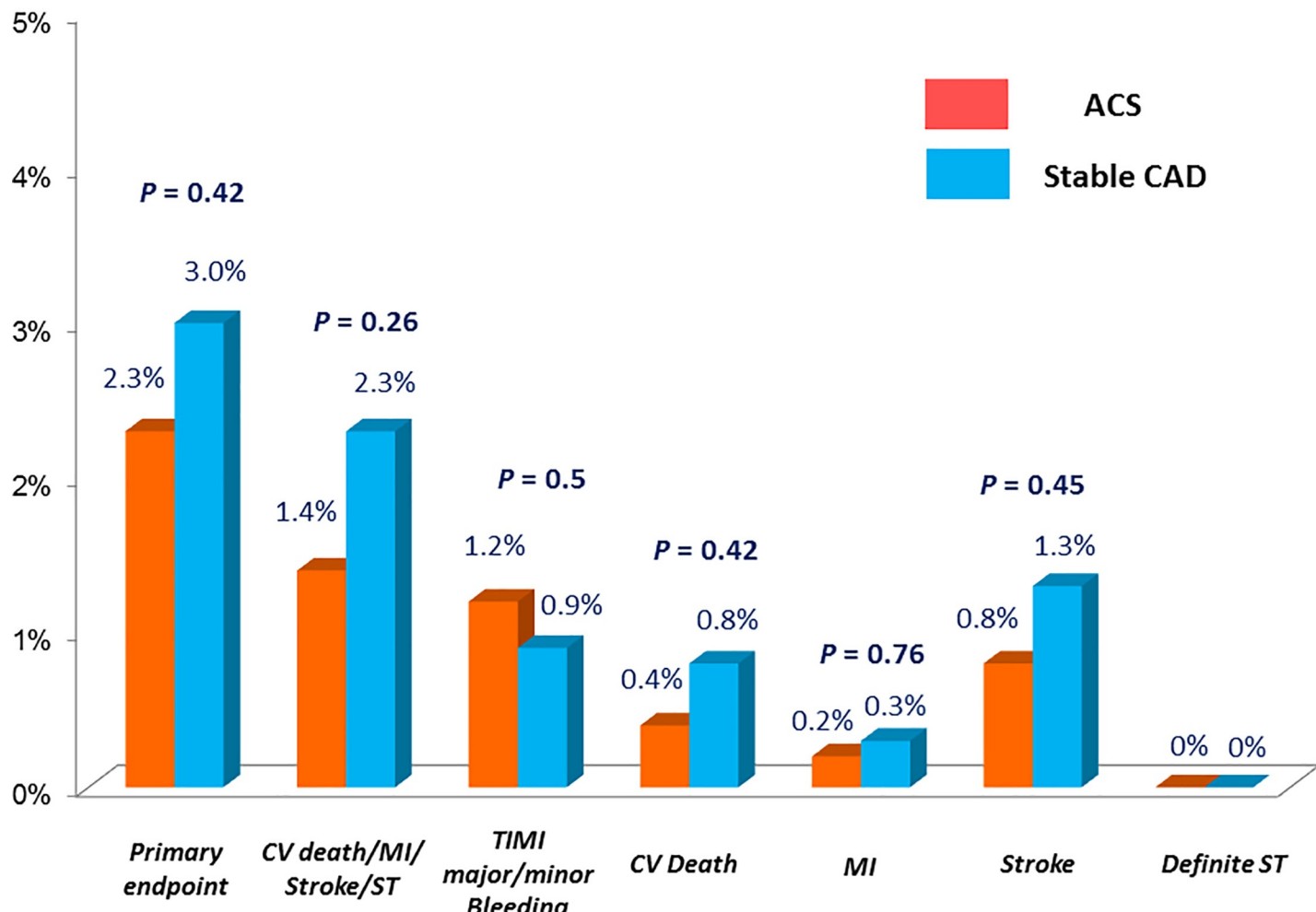

**Fig 4. Clinical outcomes at 1-year.** Event rates indicated the cumulative 1-year incidence estimated by Kaplan-Meier method. Primary endpoint was defined as a composite of CV death, MI, stroke, definite ST and TIMI major/minor bleeding. ACS, acute coronary syndrome; CAD, coronary artery disease; CV death, cardiovascular death; MI, myocardial infarction; ST, stent thrombosis; TIMI, Thrombolysis in Myocardial Infarction.

Versus Twelve Months of Dual Antiplatelet Therapy After Drug-Eluting Stent Implantation in ST-Elevation Myocardial Infarction) trial, 6-month DAPT was also non-inferior to 12-month DAPT in patients with STEMI undergoing PCI with zotarolimus-eluting stent in terms of the primary endpoint of a composite of all-cause mortality, MI, revascularization, stroke, and TIMI major bleeding at 18 months (4.8% versus 6.6%, P for non-inferiority = 0.004, and P for superiority = 0.26).[18] In contrast, in a network meta-analysis, 3-month DAPT, but not 6-month DAPT, was associated with higher rates of MI or ST as compared to 12-month DAPT in ACS patients, but not in stable CAD patients.[19] In this meta-analysis, data of the 3-month DAPT were derived from the RESET (REal Safety and Efficacy of 3-month dual anti-platelet Therapy following Endeavor zotarolimus-eluting stent implantation), and the OPTI-MIZE (Optimized Duration of Clopidogrel Therapy Following Treatment With the Zotarolimus-Eluting Stent in Real-World Clinical Practice) randomized trials. [20,21] In the

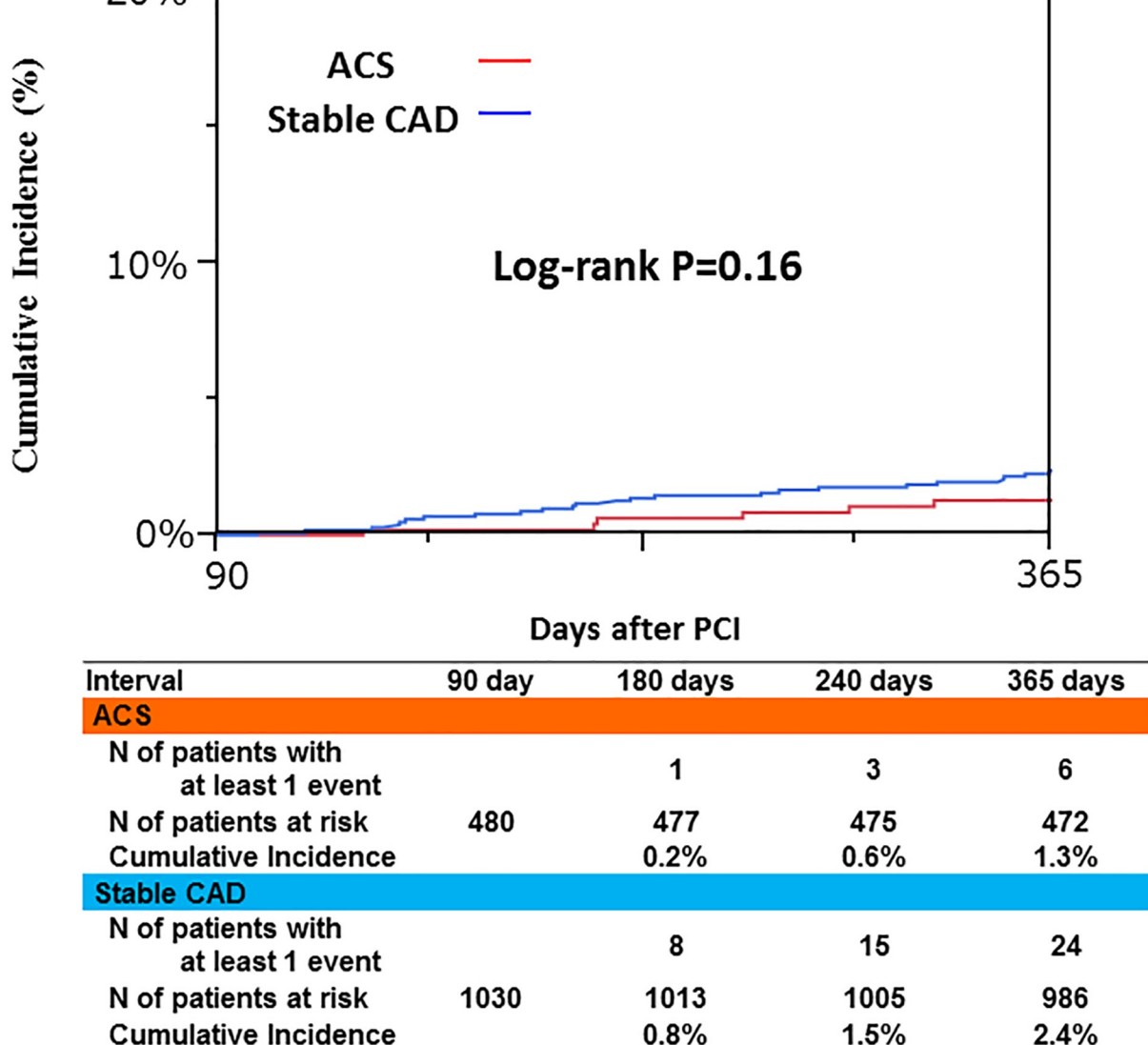

**Fig 5. Cumulative incidence of the primary endpoint between 3-month and 12-month.** Primary endpoint was defined as a composite of cardiovascular death, MI, stroke, definite ST and TIMI major/minor bleeding.

RESET and OPTIMIZE trials, 3-month DAPT was evaluated with use of Endeavor™ zotarolimus-eluting stents, which has late lumen loss similar to BMS. CoCr-EES was reported to have significantly lower rates for ST as compared with BMS in patients with STEMI in the 1-year results of the EXAMINATION (Everolimus-Eluting Stents Versus Bare-Metal Stents in ST-Segment Elevation Myocardial Infarction) trial.[22] Therefore, the efficacy of short DAPT in ACS patients should be evaluated in those patients treated with new-generation DES, CoCr-EES in particular. The GLOBAL LEADERS trial explored the efficacy of the experimental regimen of 1-month DAPT followed by ticagrelor monotherapy as compared with the standard

**Table 4. Clinical outcomes between 3-month and 12-month.**

| | No. of patients with at least one event (Cumulative incidence) | | P Value |
| --- | --- | --- | --- |
| | ACS | Stable CAD | |
| Primary Endpoint | 6 (1.3%) | 24 (2.4%) | 0.16 |
| Death | | | |
| All-cause | 2 (0.4%) | 23 (2.2%) | 0.01 |
| Cardiac death | 1 (0.2%) | 7 (0.7%) | 0.24 |
| Cardiovascular death | 1 (0.2%) | 7 (0.7%) | 0.24 |
| Non-cardiac death | 1 (0.2%) | 16 (1.6%) | 0.02 |
| Myocardial infarction | 1 (0.2%) | 1 (0.1%) | 0.59 |
| Stroke | | | |
| Any | 1 (0.2%) | 10 (1.0%) | 0.1 |
| Ischemic | 1 (0.2%) | 8 (0.8%) | 0.18 |
| Hemorrhagic | 1 (0.2%) | 2 (0.2%) | 0.96 |
| Bleeding | | | |
| TIMI major | 3 (0.6%) | 7 (0.7%) | 0.89 |
| TIMI minor/major | 4 (0.8%) | 8 (0.8%) | 0.92 |
| TIMI minimal/minor/major | 10 (2.1%) | 16 (1.6%) | 0.48 |
| GUSTO severe | 3 (0.6%) | 4 (0.4%) | 0.54 |
| GUSTO moderate/severe | 3 (0.6%) | 8 (0.8%) | 0.73 |
| Definite stent thrombosis | 0 (0%) | 0 (0%) | |
| Stent thrombosis | | | |
| Possible | 0 (0%) | 6 (0.6%) | 0.09 |
| Probable | 0 (0%) | 0 (0%) | |
| Definite or probable | 0 (0%) | 0 (0%) | |
| Definite, probable or possible | 0 (0%) | 6 (0.6%) | 0.09 |
| Death or myocardial infarction | 3 (0.6%) | 24 (2.3%) | 0.02 |
| Cardiovascular death or myocardial infarction | 2 (0.4%) | 8 (0.8%) | 0.41 |
| Cardiovascular death, MI or stroke | 3 (0.6%) | 18 (1.8%) | 0.08 |
| Cardiovascular death, MI, stroke or definite ST | 3 (0.6%) | 18 (1.8%) | 0.08 |
| Target-lesion revascularization | 8 (1.7%) | 21 (2.1%) | 0.6 |
| Target-vessel revascularization | 15 (3.1%) | 37 (3.6%) | 0.61 |
| Any coronary revascularization | 29 (6.0%) | 69 (6.8%) | 0.56 |
| Coronary-artery bypass grafting | 1 (0.2%) | 2 (0.2%) | 0.96 |

Values are expressed as number (%).

Cumulative incidence was estimated by Kaplan-Meier method.

Abbreviations are as in Table 2.

regimen of 12-month DAPT followed by aspirin monotheray in patients with stable CAD or ACS who underwent PCI with a biolimus A9-eluting stent.[23] Overall, the study failed to demonstrate the superiority of the experimental regimen for the primary endpoint of all-cause death or new Q-wave MI (rate ratio 0.87 [95% CI 0.75–1.01]; p = 0.073).[23] However in the ACS population, dropping aspirin beyond 1-month and continuing ticagrelor monotherapy was associated with a significantly lower risk for major bleeding with a trend for decreasing mortality at 1-year, suggesting that the current DAPT regimen with aspirin and ticagrelor for 1 year in ACS patients might be too intensive. [24]

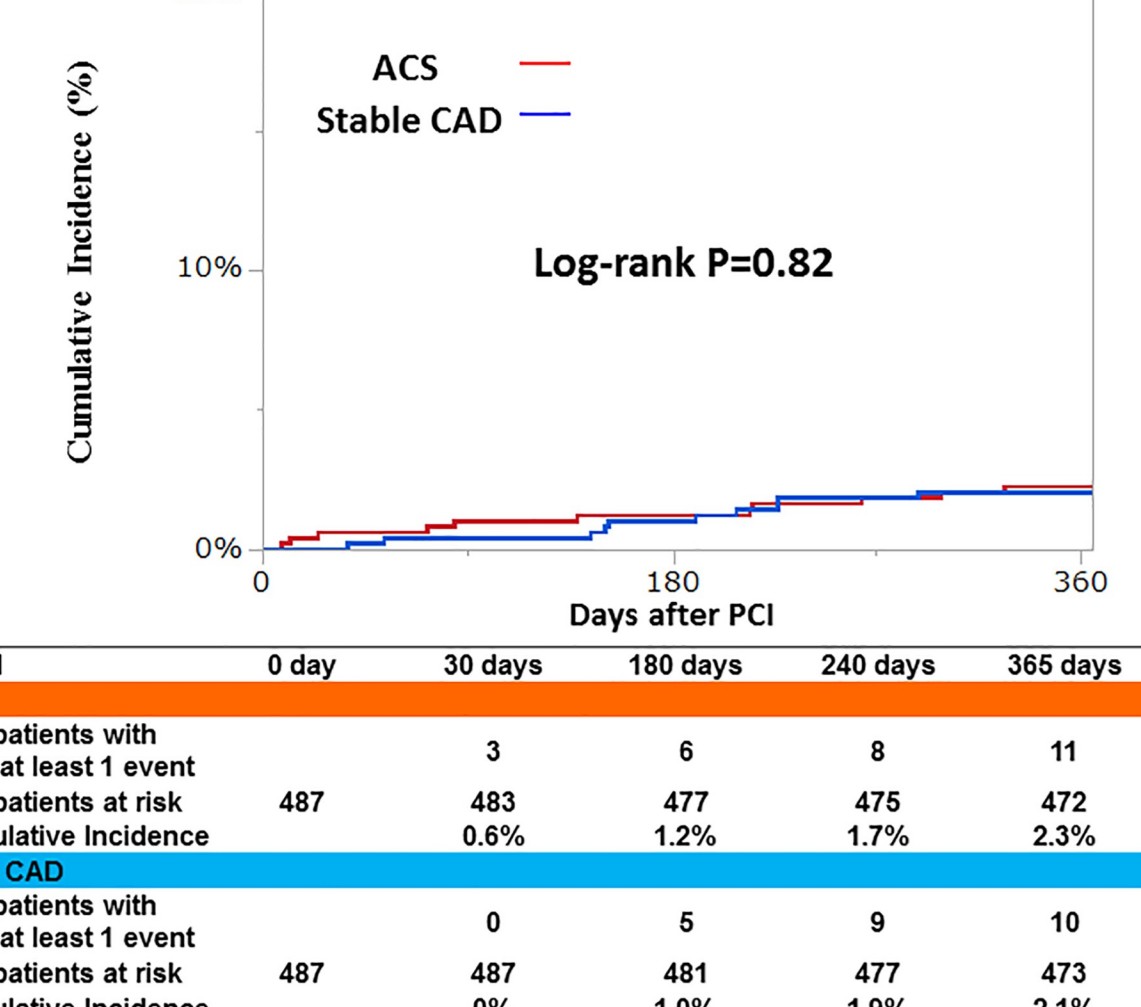

**Fig 6. Clinical outcomes at 1-year in the propensity matched cohort.** Event rates indicated the cumulative 1-year incidence estimated by Kaplan-Meier method. Primary endpoint was defined as a composite of CV death, MI, stroke, definite ST and TIMI major/minor bleeding. ACS, acute coronary syndrome; CAD, coronary artery disease; CV death, cardiovascular death; MI, myocardial infarction; ST, stent thrombosis; TIMI, Thrombolysis in Myocardial Infarction.

In the present post-hoc analysis of the STOPDAPT trial demonstrated that 3-month DAPT after CoCr-EES implantation was associated with similarly low event rate for the primary endpoint without any definite ST events in both ACS and stable CAD patients, although 47% of ACS patients had troponin-negative UA at low-risk of event recurrence. Furthermore, the STOPDAPT trial had mostly enrolled stable patients after PCI, which might be one of the reasons for the low event rate even in ACS patients. Therefore, 3-month DAPT could be an option in selected patients with ACS in the new-generation DES era. In the STOPDAPT-2 trial including approximately 40% of ACS patients, 1-month DAPT followed by clopidogrel monotherapy provided a net clinical benefit for ischemic and bleeding events over 12-month DAPT with aspirin and clopidogrel after CoCr-EES implantation.[25] The STOPDAPT-2 ACS trial

are ongoing to evaluate 1-month DAPT followed by clopidogrel monotherapy as compared with 12-month DAPT with aspirin and clopidogrel in all-comer patients undergoing PCI using CoCr-EES (ClinicalTrials.gov: NCT03462498).

## Study limitation

There are several important limitations in the current study. First, selection bias toward inclusion of patients with lower ischemic risk should be considered when interpreting the result of this study. Despite the all-comer study design, those ACS patients with large thrombotic burden and/or extensive atherosclerotic burden might well have not been included in this short DAPT trial. Indeed, patients with STEMI and NSTEMI were more often included in the non-enrolled group. Second, baseline characteristics were more complex in the stable CAD group than in the ACS group. Multivariable analysis might not be able to fully adjust the measured and unmeasured confounders. Third, patients received clopidogrel as a P2Y12 receptor blocker even in patients with ACS. Prasugrel or ticagrelor were not available in the enrollment period between 2012 and 2013 in Japan. Prasugrel or ticagrelor are generally preferred in ACS patients in the current clinical practice, especially in non-Asian patients. Fourth, as this study was conducted only in Japanese centers, the external validity to non-Asian patients would be limited. Finally, the sample size calculation was conducted for the main analysis, but not for this post-hoc analysis. Therefore, the number of patients enrolled in this study was underpowered, and not large enough to evaluate the low frequency event such as ST, especially in those with ACS.

## Conclusion

Stopping DAPT at 3 months after CoCr-EES implantation in patients with ACS including 47% of unstable angina was as safe as that in patients with stable CAD.

## Supporting information

**S1 Table. Patient characteristics: Enrolled versus non-enrolled patients.**
(DOCX)

**S2 Table. Patient characteristics in propensity matched cohort.**
(DOCX)

**S3 Table. Clinical outcomes at 12 months in propensity matched cohort.**
(DOCX)

**S1 Protocol. STOPDAPT ShorT and OPtimal duration of Dual AntiPlatelet Therapy study.**
(PDF)

**S1 Appendix.**
(DOCX)

## Acknowledgments

We deeply appreciate the following co-investigators of the participating centers: Saiseikai Kumamoto Hospital; Koichi Nakao, Shinzo Miyamoto, Kyoto University Hospital; Takeshi Kimura, Masahiro Natsuaki, Erika Yamamoto, Saitama Cardiovascular and Respiratory Center; Tetsuya Ishikawa, Joshi Tsutsumi, Chikamori Hospital; Kazuya Kawai, Shuichi Seki, Osaka City General Hospital; Kei Yunoki, Akira Itoh, Mashiko Hospital; Shogo Shimizu,

National Hospital Organization Kyoto Medical Center; Masaharu Akao, Mitsuru Ishii, Mitsubishi Kyoto Hospital; Shinji Miki, Tetsu Mizoguchi, Masashi Kato, Kimitsu Chuo Hospital; Masashi Yamamoto, Seirei Hamamatsu General Hospital; Hisayuki Okada, Nagai Hospital; Kozo Hoshino, Teine Keijinkai Hospital; Mitsugu Hirokami, Juntendo University Shizuoka Hospital; Satoru Suwa, Saiseikai Yokohamashi Tobu Hospital; Toshiya Muramatsu, Norihiro Kobayashi, Okamura Memorial Hospital; Yasuhiro Tarutani, Osaka Red Cross Hospital; Tsukasa Inada, Fujio Hayashi, Iwate Medical University Hospital; Yoshihiro Morino, Yu Ishikawa, Mie University Hospital; Masaaki Ito, Takashi Tanigawa, Toshiki Sawai, Kurashiki Central Hospital; Kazushige Kadota, Hiroyuki Tanaka, Shiga Medical Center for Adults; Shigeru Ikeguchi, Masaharu Okada, Yasutaka Inuzuka, Saiseikai Fukuoka General Hospital; Takeshi Serikawa, Toshiyuki Kozai, Masahiro Natsuaki, Mitsui Memorial Hospital; Kengo Tanabe, Takuya Hashimoto, Caress Sapporo Tokeidai Memorial Hospital; Kazushi Urasawa, Ryoji Koshida, Cardiovascular Center Hokkaido Ohno Hospital; Takehiro Yamashita, Taishi Maeno, Yokohama City University Medical Center; Kazuo Kimura, Kiyoshi Hibi, Kyoto Second Red Cross Hospital; Hiroshi Fujita, Koji Isodono, Hirakata Kohsai Hospital: Shoji Kitaguchi, Yuko Morikami, Hamamatsu Medical Center; Masakazu Kobayashi, Terumori Sato, Kobe City Medical Center General Hospital; Makoto Kinoshita, Japan Community Health Care Organization Hokkaido Hospital; Keiichi Igarashi, Jungo Furuya, Tenri Hospital; Yoshihisa Nakagawa, Toshihiro Tamura, Tokushima Red Cross Hospital; Koichi Kishi, Sakakibara Memorial Hospital; Tetsuya Tobaru, Itaru Takamisawa, Hyogo Prefectural Amagasaki Hospital; Yoshiki Takatsu, Ryoji Taniguchi, Hoshi General Hospital; Yoshitane Seino, Yasuhiro Shimizu, Kokura Memorial Hospital; Kenji Ando, Kyohei Yamaji, Takeda Hospital; Noriyuki Kinoshita, The Cardiovascular Institute Hospital; Junji Yajima, Nobuhiro Murata, Yokohama City University Hospital; Teruyasu Sugano, Hideyuki Ogawa, Masayoshi Kiyokuni, Japanese Red Cross Society Wakayama Medical Center; Takashi Tamura, Kousuke Takahashi, University of Occupational and Environmental Health Japan; Shinjo Sonoda, Kuninobu Kashiyama, Teikyo University Hospital; Hiroyuki Kyono, Fukuyama Cardiovascular Hospital; Hideo Takebayashi, Yuetsu Kikuta, National Cerebral and Cardiovascular Center Hospital; Satoshi Yasuda, Hiroki Sakamoto, Yasuhide Asaumi, Osaka City University Hospital; Minoru Yoshiyama, Takao Hasegawa, Tomokazu Iguchi, Wakayama Medical University Hospital; Takashi Akasaka, Tomoyuki Yamaguchi, Juntendo University Hospital; Katsumi Miyauchi, Shinya Okazaki, Matsue Red Cross Hospital; Kinya Shirota, Bell Land General Hospital; Toru Kataoka, Yuya Sakamoto, Kinki University Hospital; Shunichi Miyazaki, Masakazu Yasuda, Sendai Open Hospital; Atsushi Kato, Kenya Saji, Toyohashi Heart Center; Takahiko Suzuki, Yoshihisa Kinoshita, Aichi Medical University Hospital; Tetsuya Amano, Hiroaki Takashima, Saiseikai Matsuyama Hospital; Kouki Watanabe, Susumu Shigemi, Mie Heart Center; Hideo Nishikawa, Hiroyuki Suzuki, Tokyo Women's Medical University Hospital; Junichi Yamaguchi, Kazuho Kamishima, Saitama Medical Center Jichi Medical University; Junya Ako, Takuji Katayama, Wada Hiroshi, Sumitomo Hospital; Hisatoyo Hiraoka, Yuji Yasuga.

## Author Contributions

**Conceptualization:** Kazushige Kadota, Yoshihiro Morino, Keiichi Igarashi Hanaoka, Kengo Tanabe, Ken Kozuma, Takeshi Kimura.

**Formal analysis:** Masahiro Natsuaki, Takeshi Morimoto.

**Funding acquisition:** Takeshi Kimura.

**Investigation:** Masahiro Natsuaki, Erika Yamamoto, Hirotoshi Watanabe, Yutaka Furukawa, Mitsuru Abe, Koichi Nakao, Tetsuya Ishikawa, Kazuya Kawai, Kei Yunoki, Shogo Shimizu,

Masaharu Akao, Shinji Miki, Masashi Yamamoto, Hisayuki Okada, Kozo Hoshino, Kazushige Kadota, Yoshihiro Morino, Keiichi Igarashi Hanaoka, Kengo Tanabe, Ken Kozuma, Takeshi Kimura.

**Methodology:** Masahiro Natsuaki, Takeshi Kimura.

**Supervision:** Takeshi Kimura.

**Writing – original draft:** Masahiro Natsuaki.

**Writing – review & editing:** Takeshi Kimura.

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
