## [Decision Letter · Decision Letter 0]

24 Sep 2019

PONE-D-19-23933

One-year Clinical Outcomes of Patients With versus Without Acute Coronary Syndrome with 3-Month Duration of Dual Antiplatelet Therapy after Everolimus-eluting Stent Implantation

PLOS ONE

Dear Dr. Kimura,

Thank you for submitting your manuscript to PLOS ONE. After careful consideration, we feel that it has merit but does not fully meet PLOS ONE’s publication criteria as it currently stands. Therefore, we invite you to submit a revised version of the manuscript that addresses the points raised during the review process.

All reviewers have raised serious concerns about the comparison between two very different populations (ACS vs non-ACS) in terms of baseline characteristics and the about low incidence of events in the ACS group, that mostly consists of patients with unstable angina. 

Reviewer 2 has raised concerns about population selection bias, that may hamper the generalizability of these results, and especially about the need to provide non-inferiority analysis rather than a superiority analysis. 

Reviewer 4 has raised concerns about the actual differences between ACS and stable patients and suggested to adjust for residual confounders by providing a propensity matching. 

All reviewers raised concerns about the actual risk of patients with ACS as compared to stable. CAD. Indeed, reviewers underline that the risk of events was higher in stable CAD patients and that ACS appears protective at univariate analysis, that is contraintutive.  

Statistical reviewer (Reviewer 3) has requested to upload data and codes to comply with Journal’s guidelines and policy. 

I have concerns about the real need to compare two different populations from two different trials. Probably this part of the study should be omitted, else a complex propensity matching between patients from the different trials should be performed. 

We would appreciate receiving your revised manuscript by Nov 08 2019 11:59PM. To enhance the reproducibility of your results, we recommend that if applicable you deposit your laboratory protocols in protocols.io, where a protocol can be assigned its own identifier (DOI) such that it can be cited independently in the future. For instructions see: http://journals.plos.org/plosone/s/submission-guidelines#loc-laboratory-protocols

We look forward to receiving your revised manuscript.

Kind regards,

Giuseppe Andò, M.D., Ph.D.

Academic Editor

PLOS ONE

Journal Requirements:

2. Please provide additional details regarding participant consent. In the ethics statement in the Methods and online submission information, please ensure that you have specified (1) whether consent was informed and (2) what type you obtained (for instance, written or verbal, and if verbal, how it was documented and witnessed). If your study included minors, state whether you obtained consent from parents or guardians.

3. Please confirm whether your Institutional Review Board specifically approved the study.

'Abbott Vascular is the funding source of this study'              

'Takeshi Kimura, Keiichi Igarashi, Kazushige Kadota, Kengo Tanabe, Yoshihiro Morino,

and Ken Kozuma were advisory board members of Abbott Vascular.'

Additional Editor Comments (if provided):

Reviewers' comments:

Reviewer's Responses to Questions

**Comments to the Author**

1. Is the manuscript technically sound, and do the data support the conclusions?

Reviewer #1: Yes

Reviewer #2: Partly

Reviewer #3: Yes

Reviewer #4: Partly

Reviewer #5: Yes

2. Has the statistical analysis been performed appropriately and rigorously? 

Reviewer #1: Yes

Reviewer #2: No

Reviewer #3: Yes

Reviewer #4: No

Reviewer #5: I Don't Know

3. Have the authors made all data underlying the findings in their manuscript fully available?

Reviewer #1: Yes

Reviewer #2: Yes

Reviewer #3: No

Reviewer #4: Yes

Reviewer #5: No

4. Is the manuscript presented in an intelligible fashion and written in standard English?

Reviewer #1: Yes

Reviewer #2: Yes

Reviewer #3: Yes

Reviewer #4: Yes

Reviewer #5: Yes

5. Review Comments to the Author

Reviewer #1: The submitted manuscript is a post-hoc analysis of the STOPDAPT trial, investigating the safety of a 3-month DAPT regimen after successful CoCr-EES implantation in patients with or without ACS. The authors aimed at evaluating if a short DAPT regimen is similarly safe in SCAD and ACS patients since those with an acute presentation have been traditionally considered at higher risk for recurrent thrombotic events and so in need for a more prolonged (at least 1-year) DAPT. The trial has a valid scientific ground, and methods and analyses are rigorously conducted. The authors concluded that “Stopping DAPT at 3-month after CoCr-EES implantation in patients with ACS was as safe as that in patients with stable CAD.”

Detailed comments to be considered:

- In the main paper published in 2016, a sub-group analysis investigating the ACS vs. non-ACS group already showed the absence of a significant interaction between clinical presentation and outcomes. Thus, these results are only confirmatory of previously published data. However, taking into account the recognized limits of sub-group analyses, the presented data are of potential interest addressing this issue with a more rigorous and detailed approach.

- The rate of the primary endpoint events was similar in the two cohorts of patients. However, while the rate of events in the stable CAD group (3%) is in line with recent epidemiological studies, the rate of 1-year events for the ACS (2.3%) appears unusually low as compared with other reports, and less than half of the RESET cohort (4.8%) that was used as historical reference by the authors. Surprisingly, the rate of events results even higher (although only numerically) in the SCAD group compared with the ACS, which is difficult to justify biologically. This difference is even more evident at the land-mark analysis, where the risk of future events in SCAD is almost twice of that in ACS (2.4% s. 1.3%). These results might be at least in part related to the high prevalence of patients with unstable angina in the ACS group. Indeed, 1,266 patients out of 1,525 (83% of the total population) had a non-MI presentation, with only 44 patients having an NSTEMI. For this reason, no conclusions should be drawn for the ACS “troponin-positive” patients, which actually represent those at higher risk of recurrent thrombotic events and for which more concerns exist in the case of premature DAPT discontinuation. Accordingly, the present results cannot be applied to the whole ACS spectrum but limited to those at low-risk of event recurrence (comparable to those with SCAD). The authors should carefully discuss this point in the manuscript and be more cautious with their conclusions.

- It is not clear which proportion of patients received intracoronary imaging to guide PCI. In the STOPDAPT-2, the (same) investigators reported a high rate of intracoronary imaging guidance according to the clinical practice in Japan. They stated "...the vast majority of patents in this study underwent PCI guided by intracoronary imaging devices, which are rarely used in the United States and Europe. Therefore, caution is warranted in extrapolating the current study results outside of Japan". Please, clarify the percentage of patients receiving intracoronary imaging in the present analysis, also reporting data for stable CAD and ACS patients separately (if available).

- Introduction: “However, it is unclear whether the distinction between ACS and stable CAD regarding the recommendation on the DAPT duration was based on a solid scientific rationale.” This sentence should be probably reworded, as the scientific rationale for recommending a longer DAPT in patients with ACS, compared with those with stable CAD, is robust and based on the higher risk of future thrombotic events.

- “Ethics” section: The full list of the individual centers participating in the trial can be reported in the supplementary appendix.

- “Procedures” section: “Antiplatelet regimen included aspirin (≥81mg daily) …”. Which was the maximum dose of daily aspirin allowed in the trial? Did any patients receive ASA >100 mg? Please, report these data (if available).

- All patients enrolled in the trial were treated with clopidogrel. Please, briefly discuss the absence of patients receiving ticagrelor/prasugrel.

- Endpoint and definition: In the era of high-sensitivity troponin, the diagnosis of UA is becoming rarer than in the past. Please clarify in the text which cardiac biomarkers were used for distinguishing UA vs NSTEMI, and how troponin was evaluated in the study (high-sensitivity?). Were the same methods for dosing cardiac biomarkers used across the 58 centers involved in the study (or not)? This might be a potential source of confounding in a multicentre study. Moreover, the proportion of UA patients appears very high in what the authors define as an all-comers population, also if compared with the STOPDAPT-2 population. The authors report in the flow-chart of the study that 2,054 patients were excluded (actually, more than those enrolled). Can the authors clarify which was the proportion of patients with an ACS (UA, STEMI, and NSTEMI) or SCAD in the "non-enrolled" population? We can assume most of these patients were excluded because presenting with STEMI or NSTEMI and then considered by the physicians at high-risk of complications if treated with a short DAPT. This (prudent) attitude by the investigators might have biased patients enrollment, resulting in a "low-risk" study population (substantially including UA and SCAD, and excluding STEMI and NSTEMI), for which the conclusions should be applied. In this context, it would be of interest if the authors could provide the GRACE score of the NSTE-ACS patients.

- Statistical analysis: “Sample size calculation of this study was previously described.” The sample size was previously described for the main analysis, and not for this post-hoc analysis. Conversely, the present analysis seems to be underpowered and non-conclusive for its purpose. Please clarify.

- How were the patients on oral anticoagulants managed? Did the duration of DAPT in this subgroup differ from the rest of the population?

- Conclusions: should be extended, and reworded to clarify that most of ACS patients were UA.

- Please, better define which medications were considered among "strong statins".

- Please, diffusely correct typos in the text: i.e., "Statins use" instead of "Stains use" in Table 3.

Reviewer #2: In this study the authors aim at comparing the outcome in patients with ACS versus stable CAD undergoing PCI and subsequent double antiplatelet therapy (DAPT) with clopidogrel plus aspirin) for 3 months, followed by single antiplatelet therapy (SAPT) with aspirin for 12 months. Furthermore, the authors compared the clinical outcomes of the STOPDAPT trial with the RESET trial in both ACS and stable CAD patients. The authors concluded that a 3-month DAPT in ACS patients is as safe as in stable CAD patients.

The aim of this study is interesting; however, I have several comments to the authors.

Major comments:

• The authors stated that, among candidate patients, only those judged eligible by the physician were enrolled. In fact, among 3580 candidate patients, 2054 (far more than the half) were not enrolled. As the authors stated in the limitation section, this leads to a selection bias, that should be more clearly explained by the authors in discussing the results. Furthermore, as in a previous paper of the same trial (DOI 10.1007/s12928-015-0366-9), the main characteristics of not enrolled patients should be presented (also in appendix) or, at east, cited.

• Although the eligibility criteria of the RESET were comparable to those of the STOPDAPT, in the former study all patients with previous DES were excluded. Considering that, in RESET study, more patients that in STOPDAPT had previous PCI, a consistent number of patients in this study had a previous BMS implanted. Furthermore, in STOPDAPT only those judged eligible by the physician were enrolled. Importantly the baseline characteristics of the population of the two trials are strongly different (as shown in table s1), suggesting a higher risk profile in RESET population (more diabetics, more end stage CKD, more MVD…). This suggest that many confounders could bias the results and an adjustment through a propensity score should be performed before comparing the outcome of this two populations.

• There are significant differences between the ACS and stable CAD group in terms of many important variables (more AF, PAD, previous MI, Previous PCI/CABG, MVD in stable CAD group). Since these large differences between the two arms could be not adequately adjusted by a multivariable analysis, an adjustment throughout a propensity score would be a better option.

• The fact that in stable CAD 6 possible stent thrombosis occurred against 0 in ACS group clearly reflect a difference among the two populations. The same is for the difference, in terms death ((particularly non-cardiac death), between ACS and CAD at the landmark analysis. It is difficult to believe that that chronic CAD patients had a worse outcome.

• If the aim of the study is to demonstrate that, after an ACS, a 3-month DAPT after CoCr-EES implantation in patients with ACS is as safe as (or not less safe than) in stable CAD patients, a non-inferiority analysis should be planned. In fact, the non-significance at a superiority analysis cannot be used to demonstrate that there is not any significant difference between the two arms nor that an arm is not inferior to a second arm. This is particularly true for the endpoint for which the study is underpowered (e.g. stent thrombosis).

• It should be discussed that, for this study, the external validity to non-Asian patients is limited firstly because the study was conducted only in Japanese centres, secondly because the main P2Y12 inhibitor used was Clopidogrel (whereas ticagrelor or prasugrel should be generally preferred, especially in non-Asian patients)

Minor comments:

• In line 268, why do the authors use the plural form for STOPDAPT-2 ACS trial?

• I think that hyphens are sometimes incorrectly used. For example, in conclusion section, “Stopping DAPT at 3-month” should be replaced with “Stopping DAPT at 3 months”.

Reviewer #3: The manuscript addresses an interesting topic. The data are of certain interest and the employed methods are sound. The results might be very useful for future researchers.

Some comments follow.

1. Data are not fully available. This is not in line with the journal's guidelines. Please, update the data and the code used to obtain parameters estimates. This would ensure the reproducibility of the results and allow the reviewers to check for the appropriateness of the methods.

2. My main doubts concern some aspects of the modelling.

2a. Firstly, the authors must check for heterogeneity between centers. A major, and quite strong, assumption in the modelling is that patients belonging to different centers are homogeneous, i.e. no differences arise between multiple centers. Please, provide an extension of the modelling that takes into account for the presence of heterogeneity between centers. A random effects model may be an option.

2b. The use of Cox survival modelling is sound. Nevertheless, several assumptions must be fulfilled to ensure a proper statistical inference. It is desirable to determine whether a fitted Cox regression model adequately describes the data. I will briefly consider three kinds of diagnostics: for violation of the assumption of proportional hazards; for influential data; and for nonlinearity in the relationship between the log hazard and the covariates. Please, provide evidence that model assumptions are fulfilled or, if thery not, modify the model specification accordingly.

2c. As minor points: please explain the difference between adjusted and non-adjusted coefficients; furthermore, please consider a model selection procedure (e.g. LASSO) to identify the relevant variables.

Reviewer #4: - In consideration of the relevant clinical and angiographic differences between the two compared groups, you have to perform a propensity pair-matched analysis, to increase accuracy of your results. Your current adjusted analysis is very limited and you have the number to perform a better analysis.

- Events tended to be higher in the stable CAD groups, please provides analysis to better understand this greater risk.

- You have to better explain what are the scientific and practical implications of your data and findings. Try to reinforce the overall meaning of your data.

Reviewer #5: It is unclear how events occurred more often in the Stable CAD rather than in the ACS arm. ACS patients including NSTEMI and STEMI patients have a significantly higher risk of ischemic and bleeding complication compared to stable patients. This point should be thoroughly evaluated as it hampers to draft conclusion from this data. In fact ACS at univariate analysis seems tendentially protective from adverse events, which tend to lessen after adjustment. Ultimately it is not clear to this reviewer the comparison between ACS and SCAD for clinical events considering that due to more conservative patient selection in the ACS arm (see Table 1). STOPDAPT as a single arm study cannot either give information regarding the impact of different terms of DAPT duration.

6. PLOS authors have the option to publish the peer review history of their article (what does this mean?). If published, this will include your full peer review and any attached files.

Reviewer #1: Yes: Felice Gragnano

Reviewer #2: No

Reviewer #3: No

Reviewer #4: No

Reviewer #5: No

---

## [Author Response · Author response to Decision Letter 0]

17 Nov 2019

Responses to the editor’s comments

We appreciate the kind editing and comments from the editor. 

We amended the manuscript as suggested by the editor. 

Editor: 

I have concerns about the real need to compare two different populations from two different trials. Probably this part of the study should be omitted, else a complex propensity matching between patients from the different trials should be performed.

We appreciate the comments.

As the editor suggested, we omitted the comparison of two different populations from the STOPDAPT and RESET trials.

---

## [Decision Letter · Decision Letter 1]

3 Dec 2019

PONE-D-19-23933R1

One-year Clinical Outcomes of Patients With versus Without Acute Coronary Syndrome with 3-Month Duration of Dual Antiplatelet Therapy after Everolimus-eluting Stent Implantation

PLOS ONE

Dear Dr. Kimura,

Thank you for submitting your manuscript to PLOS ONE. After careful consideration, we feel that it has merit but does not fully meet PLOS ONE’s publication criteria as it currently stands. Therefore, we invite you to submit a revised version of the manuscript that addresses the points raised during the review process.

All reviewers appreciated your revised versions of the manuscript. While clinical reviewers do not have further  concerns, statistical reviewer asks to address a residual comment about the Cox model. Importantly, this comment will not influence the final decision to accept this manuscript.  

We would appreciate receiving your revised manuscript by Jan 17 2020 11:59PM. To enhance the reproducibility of your results, we recommend that if applicable you deposit your laboratory protocols in protocols.io, where a protocol can be assigned its own identifier (DOI) such that it can be cited independently in the future. For instructions see: http://journals.plos.org/plosone/s/submission-guidelines#loc-laboratory-protocols

We look forward to receiving your revised manuscript.

Kind regards,

Giuseppe Andò, M.D., Ph.D.

Academic Editor

PLOS ONE

Reviewers' comments:

Reviewer's Responses to Questions

**Comments to the Author**

1. If the authors have adequately addressed your comments raised in a previous round of review and you feel that this manuscript is now acceptable for publication, you may indicate that here to bypass the “Comments to the Author” section, enter your conflict of interest statement in the “Confidential to Editor” section, and submit your "Accept" recommendation.

Reviewer #1: All comments have been addressed

Reviewer #2: All comments have been addressed

Reviewer #3: (No Response)

Reviewer #4: All comments have been addressed

2. Is the manuscript technically sound, and do the data support the conclusions?

Reviewer #1: Yes

Reviewer #2: Yes

Reviewer #3: Yes

Reviewer #4: Yes

3. Has the statistical analysis been performed appropriately and rigorously? 

Reviewer #1: Yes

Reviewer #2: Yes

Reviewer #3: Yes

Reviewer #4: Yes

4. Have the authors made all data underlying the findings in their manuscript fully available?

Reviewer #1: No

Reviewer #2: Yes

Reviewer #3: No

Reviewer #4: Yes

5. Is the manuscript presented in an intelligible fashion and written in standard English?

Reviewer #1: Yes

Reviewer #2: Yes

Reviewer #3: Yes

Reviewer #4: Yes

6. Review Comments to the Author

Reviewer #1: The authors addressed appropriately all the comments in the manuscript. No further comments are needed.

Reviewer #2: All my comments have been addressed.

Reviewer #3: Thank you very for having addressed most of the comments I raised.

One point needs still to be addressed. Influential observations and nonlinearities in the Cox model should be analysed. Yhe martingale residuals may be plotted against covariates to detect nonlinearity, and may

also be used to form component-plus-residual (or partial-residual) plots, again in the manner of linear and

generalized linear models. A matrix of estimated changes in the regression coefficients upon deleting each observation in turn may be used to detect influential observations.

Reviewer #4: Reviewer comments were addressed and the manuscript has been improved. I have no further comments.

7. PLOS authors have the option to publish the peer review history of their article (what does this mean?). If published, this will include your full peer review and any attached files.

Reviewer #1: No

Reviewer #2: Yes: Marco Di Maio

Reviewer #3: No

Reviewer #4: No

---

## [Author Response · Author response to Decision Letter 1]

18 Dec 2019

Responses to the reviewer’s comments

We appreciate the kind editing and comments from the editor and the reviewers. 

Reviewer #3: 

One point needs still to be addressed. Influential observations and nonlinearities in the Cox model should be analyzed. Yhe martingale residuals may be plotted against covariates to detect nonlinearity, and may also be used to form component-plus-residual (or partial-residual) plots, again in the manner of linear and generalized linear models. A matrix of estimated changes in the regression coefficients upon deleting each observation in turn may be used to detect influential observations.

Thank you very much for your comment. 

As reviewer commented, influential observations and nonlinearities in the Cox model could be analyzed. Proportional hazard assumptions for the risk-adjusting variables were assessed on the plots of log (time) versus log [-log (survival)] stratified by the variable. The assumptions were verified to be acceptable for all the variables. We thus here show the plots for age, atrial fibrillation (AF), and peripheral vascular disease (PVD) because these variables had smaller p-values. 

The main comparison of this study is ACS versus non-ACS patients, and thus other covariates are used for adjustment in the multivariate models and not relevant to the main results of this study. Therefore, the detailed inspections for residuals or linearity are not considered relevant to the manuscript and they do not affect on our results. However, we are happy to conduct these analyses if the editors and reviewers considered it necessary to accept our manuscript.

---

## [Editor Report · Decision Letter 2]

26 Dec 2019

One-year Clinical Outcomes of Patients With versus Without Acute Coronary Syndrome with 3-Month Duration of Dual Antiplatelet Therapy after Everolimus-eluting Stent Implantation

PONE-D-19-23933R2

Dear Dr. Kimura,

We are pleased to inform you that your manuscript has been judged scientifically suitable for publication and will be formally accepted for publication once it complies with all outstanding technical requirements.

With kind regards,

Giuseppe Andò, M.D., Ph.D.

Academic Editor

PLOS ONE
---

## [Editor Report · Acceptance letter]

12 Feb 2020

PONE-D-19-23933R2 

One-year Clinical Outcomes of Patients With versus Without Acute Coronary Syndrome with 3-Month Duration of Dual Antiplatelet Therapy after Everolimus-eluting Stent Implantation 

Dear Dr. Kimura:

I am pleased to inform you that your manuscript has been deemed suitable for publication in PLOS ONE. Congratulations! Your manuscript is now with our production department. 

With kind regards,

on behalf of

Dr. Giuseppe Andò 

Academic Editor

PLOS ONE